# Prediction of the Short-Term Therapeutic Effect of Anti-VEGF Therapy for Diabetic Macular Edema Using a Generative Adversarial Network with OCT Images

**DOI:** 10.3390/jcm11102878

**Published:** 2022-05-19

**Authors:** Fabao Xu, Shaopeng Liu, Yifan Xiang, Jiaming Hong, Jiawei Wang, Zheyi Shao, Rui Zhang, Wenjuan Zhao, Xuechen Yu, Zhiwen Li, Xueying Yang, Yanshuang Geng, Chunyan Xiao, Min Wei, Weibin Zhai, Ying Zhang, Shaopeng Wang, Jianqiao Li

**Affiliations:** 1Department of Ophthalmology, Qilu Hospital, Cheeloo College of Medicine, Shandong University, Jinan 250012, China; xufabao09591@qiluhospital.com (F.X.); jiaweieye@sdu.edu.cn (J.W.); 201935693@mail.sdu.edu.cn (Z.S.); ruizhang1991@yahoo.com (R.Z.); zhaowenjuan2007@163.com (W.Z.); yuxuechen2022@163.com (X.Y.); hiwen15621623898@163.com (Z.L.); 15863780685@163.com (X.Y.); gysazc@163.com (Y.G.); 18562158847@163.com (C.X.); 18764071608@163.com (M.W.); jvbin1991@163.com (W.Z.); zhangyingzy2007@126.com (Y.Z.); 2School of Computer Science, Guangdong Polytechnic Normal University, Guangzhou 510665, China; l-shaopeng@live.cn; 3State Key Laboratory of Ophthalmology, Zhongshan Ophthalmic Center, Sun Yat-sen University, Guangzhou 510085, China; xiangyf3@mail2.sysu.edu.cn; 4School of Medical Information Engineering, Guangzhou University of Chinese Medicine, Guangzhou 510182, China; jmhong@outlook.com; 5Zibo Central Hospital, Binzhou Medical University, Zibo 256603, China; wangshaopeng@medmail.com.cn

**Keywords:** deep learning, diabetic macular edema, generative adversarial networks, optical coherence tomography

## Abstract

Purpose: To generate and evaluate individualized post-therapeutic optical coherence tomography (OCT) images that could predict the short-term response of anti-vascular endothelial growth factor (VEGF) therapy for diabetic macular edema (DME) based on pre-therapeutic images using generative adversarial network (GAN). Methods: Real-world imaging data were collected at the Department of Ophthalmology, Qilu Hospital. A total of 561 pairs of pre-therapeutic and post-therapeutic OCT images of patients with DME were retrospectively included in the training set, 71 pre-therapeutic OCT images were included in the validation set, and their corresponding post-therapeutic OCT images were used to evaluate the synthetic images. A pix2pixHD method was adopted to predict post-therapeutic OCT images in DME patients that received anti-VEGF therapy. The quality and similarity of synthetic OCT images were evaluated independently by a screening experiment and an evaluation experiment. Results: The post-therapeutic OCT images generated by the GAN model based on big data were comparable to the actual images, and the response of edema resorption was also close to the ground truth. Most synthetic images (65/71) were difficult to differentiate from the actual OCT images by retinal specialists. The mean absolute error (MAE) of the central macular thickness (CMT) between the synthetic OCT images and the actual images was 24.51 ± 18.56 μm. Conclusions: The application of GAN can objectively demonstrate the individual short-term response of anti-VEGF therapy one month in advance based on OCT images with high accuracy, which could potentially help to improve treatment compliance of DME patients, identify patients who are not responding well to treatment and optimize the treatment program.

## 1. Introduction

Diabetes mellitus is a major global health epidemic, and diabetic retinopathy (DR) is well-established as the leading cause of blindness in adults, affecting one-third of diabetic patients [1,2,3]. Diabetic macular edema (DME) is one of the most critical complications of DR, which affects 7% of DR patients and can cause severe vision loss [4]. DME is characterized as the thickening and cystoid edema of the macula [2]. Focal edema is generally caused by leakage from retinal capillary microaneurysms and is often associated with surrounding rings of lipoprotein deposits in the retina, known as hard exudates [5,6]. Importantly, early diagnosis and intervention can protect the macula from irreversible functional damage and improve the prognosis of DME patients [7,8].

An increasing body of evidence suggests that intravitreally injected anti-vascular endothelial growth factor (VEGF) can achieve significant therapeutic effects in treating visual impairment related to DME [1,5,9]. In clinical practice, anti-VEGF therapy remains the first-line treatment for DME [8,9,10]. Most patients require repeated anti-VEGF injections to achieve optimal outcomes during DME management [11,12]. However, there are significant individual differences in response to anti-VEGF therapy. Accordingly, it remains challenging to predict the structural and functional changes after a single dose of anti-VEGF injection based on clinician experience, limiting the treatment plan and application [10,11,12].

A generative adversarial network (GAN) is an artificial intelligence-based “image-to-image” algorithm, which Ian Goodfellow first proposed in 2014, that can synthesize new images based on existing ones. The goal of GAN is to generate fake images that closely resemble the actual images. Over the past few years, GAN-based algorithms have yielded satisfactory results when used to predict the effect of anti-VEGF or laser therapies for neovascular age-related macular degeneration (nAMD) and central serous chorioretinopathy (CSC). The present research sought to establish an intelligent model to predict post-therapeutic structural outcomes based on pre-therapeutic optical coherence tomography (OCT) imaging with GAN. The intelligent model could predict post-therapeutic retinal characteristics, especially macula features, in advance to assist ophthalmologists in evaluating the responses to anti-VEGF therapy and arranging the treatment plan, improving clinical efficiency and visual prognosis.

## 2. Methods

### 2.1. Clinical Data and Imaging Examinations

To generate individualized post-therapeutic OCT images that could predict the short-term response of anti-VEGF injection based on pre-therapeutic images using a GAN, we retrospectively reviewed the records of patients with DME who underwent intravitreal injection of anti-VEGF therapy at the Department of Ophthalmology, Qilu Hospital, Shandong University from October 2018 to May 2021. The inclusion criteria consisted of (1) patients aged ≥18 years; (2) patients with a pre-operative diagnosis of DME based on fundus fluorescein angiography (FFA) and OCT; (3) patients treated with an injection of anti-VEGF therapy including conbercept or ranibizumab at any phase in the treatment protocol of three consecutive monthly injections and pro re nata (PRN) injections; and (4) OCT B-scan results showing macular edema. The exclusion criteria were as follows: (1) presence of any other chorioretinal diseases, including age-related macular degeneration (AMD) and polypoidal choroidal vasculopathy (PCV); (2) low image quality caused by media opacities or an abnormal signal strength index on the OCT images; and (3) OCT B-scan results showing no macular edema in the 21-line mode pattern. The follow-up points were at one month after anti-VEGF therapy. However, it was difficult to perform follow-up visits at a fixed date due to the different work schedules of DME patients. Thus, we determined a time range to ensure the study’s accuracy: 1 month ± 5 days after anti-VEGF therapy. All OCT images enrolled were stripped of personally identifiable information. The need for written informed consent was waived by our ethics committee due to the retrospective nature of the study, and all data used were fully anonymized. Moreover, this study adhered to the tenets of the Declaration of Helsinki and was approved by the institutional review board of Qilu Hospital (Ethical Code: 2021 (068)).

### 2.2. Data Collection

Pre-therapeutic and post-therapeutic B-scan OCT (Carl Zeiss Meditec. Inc., Dublin, OH, USA) images were obtained with the follow-up mode, which enabled the paired pre-therapeutic and post-therapeutic images to be scanned at the same location. The HD 21-Line was adopted as the main scan mode; this scan mode generates 21 high-definition horizontal scan lines at a depth of 2.0 mm with 8 acquired B-scans for each line, and each B-scan is composed of 1024 A-scans. The scan has an adjustable line length of 3, 6, or 9 mm. In the present study, we used a length of 9 mm and a width of 6 mm. To make optimal use of the image resources of patients with DME, the OCT images at different layers obtained from the same patient with macular edema were all included in the study (Figure 1). The OCT images were paired based on the position of the fovea and retinal vessels. The original resolution was 1264 × 596 pixels, which was first cropped into 760 × 490 pixels, and further resized into 512 × 512 pixels for input in the model training and validation process.

The whole B-scan OCT image pairs were randomly separated into two subsets, i.e., the training and validation sets. To be concrete, 561 pairs of images were randomly selected from the whole set to form the training set, while the remaining 71 image pairs were reserved to form the test set for model evaluation.

### 2.3. Image Synthesis

The state-of-the-art generative adversarial network (GAN) algorithms in deep learning were adopted to generate post-therapeutic OCT images based on pre-therapeutic OCT images [13,14]. Among existing GAN algorithms, the pix2pixHD algorithm is well-established as outperforming other competitive algorithms in many cases and was selected as our main algorithm in the training process. In addition, the comparison of pix2pixHD against other state-of-the-art algorithms such as pix2pix and CRN was included [13,14,15]. Roughly speaking, the pix2pixHD algorithm yields a high-resolution image-generating neural network in a game-playing manner. Besides the target generating network, the pix2pixHD algorithm also trained a discriminative network during the training process (Figure 2). These two networks iteratively challenge each other in the following way: the generating network tries to synthesize photo-realistic post-therapeutic images, while the discriminative network discriminates the generated images from the real ones (Figure 3). The two networks iteratively update themselves in such a competitive process, and the generating network can output synthesized images with high quality when the training process is terminated. Compared with other GAN-style algorithms, pix2pixHD also possesses a fine-designed network, enabling it to produce more photo-realistic quality images, making it more attractive for our purpose in this project. During the training process, we used Adam as the optimizer and set the momentum term of Adam to be 0.5. We also set the learning rate to 0.0002, the number of iterations to 150, and the number of iterations to linearly decay the learning rate to zero to be 150, respectively. The training process is shown in Figure 2. We implemented the above pix2pixHD model with the Python and PyTorch framework and python (V.3.5). All experiments were performed with Ubuntu 16.04 LTS and GeForce GTX 2080 Ti hardware.

### 2.4. Evaluation of Post-Therapeutic OCT Prediction Models

To qualitatively evaluate the performance of the GAN models, the quality and similarity of synthetic OCT images were evaluated independently by a screening experiment. The screening experiment involved evaluating the quality and similarity of synthetic post-therapeutic OCT images of patients with DME. All synthetic images and corresponding real OCT images were presented to two retinal specialists. They independently answered two questions: (1) is the synthetic image of sufficient quality? and (2) can you identify the synthetic image, A, or B? Only the synthetic images which were difficult to distinguish from originals with sufficient quality were further analyzed in the evaluation experiment.

To quantitatively evaluate the performance of the GAN models, we used the mean absolute error (MAE) as the evaluation metric. The MAE is calculated as the average value of the absolute error of the prediction results, which directly reflects the deviation of the predicted values from the actual values. The formula for the MAE is as follows:$$\mathrm{MAE}=\frac{1}{N}\sum_{i=1}^{N}\left|{\tilde{y}}_{i}-y_{i}\right|$$

We measured the CMT of both the synthetic and the actual images and then calculated the MAE. The CMT was measured as the outer surface of the line formed by the RPE to the outer surface of the retinal nerve fiber layer in the macular area. To ensure consistency of the measurement tool, the assessment of CMT of the synthetic images and the ground truth was conducted manually using the ImageJ software (National Institutes of Health, Bethesda, MD, USA).

The evaluation processes are shown in Figure 3.

## 3. Results

### 3.1. Demographic Data of Training Data and Testing Data

A total of 561 pairs of OCT images from 96 patients were assigned to the training set, and 71 pairs of OCT images from 21 patients were assigned to the validation set. In the training set, the patients’ mean (±SD) age was 58.57 (±9.14) years, and 48.96% were eyes from female patients. In the test set, the mean (SD) age of patients was 56.57 (±10.11) years, and 52.38% were eyes from female patients. More baseline clinical characteristics are shown in Table 1. There was no significant difference in the age, gender, VA, injection phase, anti-VEGF agent, and classification of macular edema between training and validation sets at the baseline. A total of 71 synthetic post-therapeutic images were generated based on the pre-therapeutic OCT images of patients with DME in the validation set.

### 3.2. Screening Experiment of Synthetic Images

A total of 71 synthetic OCT images were generated based on pre-therapeutic OCT images by the GAN models. The post-therapeutic OCT images predicted by the GAN models were compared with the ground truth. The pix2pixHD algorithm exhibited the highest accuracy in the predictions, outperforming the pix2pix and CRN (Table 2). Therefore, all subsequent analyses were conducted based on the pix2pixHD algorithm. During the screening experiment, 2 pairs of synthetic images considered to be inadequate by specialist 1 (J Wang) were reported to be adequate by specialist 2 (F Xu). Finally, the third specialist (J Li) was consulted, and the images were deemed to be of sufficient quality. In the following experiment designed to distinguish the synthetic OCT images from the actual images (ground truth), specialist 1 accurately identified 6 pairs of synthetic images, and specialist 2 accurately identified 4 pairs of synthetic OCT images. Most of the synthetic images (65/71) were difficult for retinal specialists to differentiate. All synthetic images that could not be identified were further analyzed in the following evaluation experiment. Examples of unqualified and distinguishable synthetic images are shown in Figure 4.

### 3.3. Evaluation Experiment of Indistinguishable Images

Two retinal specialists (Z. Shao and Y. Xiang) independently measured the CMT of all synthetic post-therapeutic OCT images during the evaluation experiment. The workflow of our study was shown in Figure 5. The mean values of the two measurements were calculated for further analysis. The evaluation experiment included 65 synthetic images output from the GAN models, with an MAE of 24.51 ± 18.56 μm. An illustration of the synthetic OCT images with different types of macular edema is shown in Figure 6. During a subgroup analysis based on the injection phase, the MAE of post-therapeutic OCT images from patients in the loading and PRN phase were 25.76 ± 20.25 μm and 23.11 ± 17.79 μm, respectively. A subgroup analysis of anti-VEGF agents showed that the MAE of post-therapeutic OCT images from patients treated with ranibizumab was 26.78 ± 19.34 μm, and the MAE of post-therapeutic OCT images from patients treated with conbercept was 22.39 ± 18.36 μm. In addition, a subgroup analysis based on different macular edema classifications showed that the MAE of post-therapeutic OCT images from patients with diffuse retinal thickening, cystoid macular edema and serous retinal detachment was 22.11 ± 18.47 μm, 32.45 ± 23.15 μm and 23.87 ± 21.65 μm, respectively. The MAEs of CMT between the synthetic OCT images and ground truth are shown in Table 3 in detail.

## 4. Discussion

To the best of our knowledge, this is the first study to predict post-anti-VEGF therapy OCT images in patients with DME by GAN. In the current study, we presented and evaluated an AI-based method to generate synthetic OCT images to predict structural alterations after anti-VEGF therapy. Importantly, more than ninety percent of the synthetic images could not be distinguished from the actual images by retinal specialists. In addition, the MAEs of the CMT between synthetic and actual OCT images were approximately 20–30 μm at 1-month prediction. Accordingly, the synthetic OCT images can accurately mirror the prognosis of patients after receiving anti-VEGF therapy.

During clinical practice, the evaluation of DME’s progression and therapeutic strategy is mainly based on OCT images, which enables quantitative analysis of the severity of macular edema and changes in the retinal structure of DME patients during treatment [3,16,17]. However, there are significant individual differences in therapeutic response in DME patients, and it can be challenging for clinicians to predict the short-term prognosis of structure and function after a single dose of different anti-VEGF drugs on experience [5,7,17]. Therefore, the application of AI technology to predict the prognosis of the retina of DME patients after treatment is expected to objectively display the therapeutic effects of different types of anti-VEGF drugs for the same DME patient through OCT images, providing a reference for clinicians to make therapeutic decisions. According to the results, our model can be applied to real-world patients during the loading phase and PRN phase, yielding comparable performance in different anti-VEGF medications such as ranibizumab and conbercept. Furthermore, the predictive performance for macular edema, including diffuse retinal thickening (DRT) and serous retinal detachment (SRD), was satisfactory. However, the predictive performance for cystoid macular edema (CME) was relatively unsatisfactory, probably due to the insensitivity of CME patients to anti-VEGF drugs and the sample limitations of our dataset.

Structural recoveries after anti-VEGF therapy are essential for ophthalmologists to assess the prognosis of visual function. Clinically, OCT images can provide more details of the retinal structure and function than other examinations for DME patients, and the CMT changes before and after therapy can directly reflect the therapeutic effect [6,16,18]. This is the first research to achieve structural prediction of the retina for DME patients. Several approaches have been implemented, part of which use baseline factors instead of visual acuity and central thickness to evaluate the therapeutic effect of DME [11,16,18,19]. According to the published reports, researchers have attempted to analyze the anti-VEGF effect in patients with different types of DME based on OCT images [1,11,17,20]. Associated prospective or retrospective studies carefully analyzed the prognostic factors of DME patients, providing vital information for clinical decision-making, but did not reveal the structural or functional detailed prognosis of DME patients after treatment. In 2021, Yu et al. accurately predicted vision and CMT after anti-VEGF treatment by incorporating DME patients’ clinical information and OCT data, which is a significant attempt to predict the prognosis of fundus diseases with artificial intelligence [21]. In our study, OCT images of patients after anti-VEGF therapy were directly obtained, which objectively demonstrated the effect of anti-VEGF therapy on patients.

During the process of model construction of post-therapeutic OCT images, we selected and successfully deployed the appropriate GAN algorithms, which fulfilled our initial purpose of conducting this research [13,14]. As discussed above, one of the main priorities of this study is to predict post-therapeutic retinal characteristics, especially macular structure. Therefore, generating OCT images with clear details is the key to success. To this end, the pix2pixHD algorithm was selected to establish an artificial intelligence-based model due to its advantages in producing high-resolution and near-realistic images [13]. As a state-of-the-art GAN-style algorithm, pix2pixHD uses a game-playing manner to iteratively achieve the optimal image generator (similar to the traditional GAN algorithms) and designs a new multi-scale generator and discriminator architectures to facilitate synthesizing high-resolution and photo-realistic images in a coarse-to-fine manner. As demonstrated in Figure 5 and comparison results in Table 2, our established pix2pixHD model successfully generated post-therapeutic images close to the ground truth, providing reliable information to help ophthalmologists predict and evaluate the short-term therapeutic response of DME patients to anti-VEGF therapy.

There were several limitations to our study. First of all, the sample size included in this study was relatively small, which may have influenced the predictive performance of DME patients. Greater sample sizes are conducive to improving the stability of the model. Moreover, the patients in our study were from the same hospital, and the training set and validation set were divided according to chronological order. Indeed, data from more centers are necessary to improve the general applicability of our model. We included OCT images at different retina levels, which expanded the sample size of OCT images but limited the sensitivity to the real-world data. Last but not least, synthetic images cannot replace the need for actual images and are only used for prediction. Indeed, postoperative OCT images are still needed for clinical evaluation.

In conclusion, our model yielded a good performance for predicting retinal morphological changes in DME patients after anti-VEGF therapy. At present, predicting therapeutic response to anti-VEGF treatment remains challenging in clinical practice. Our GAN model can accurately predict post-therapeutic OCT images one month after the anti-VEGF therapy, providing an accurate display of the postoperative OCT morphology and more prognostic information, improving treatment adherence and potentially improving the prognosis of DME patients.

## Figures and Tables

**Figure 1 jcm-11-02878-f001:**
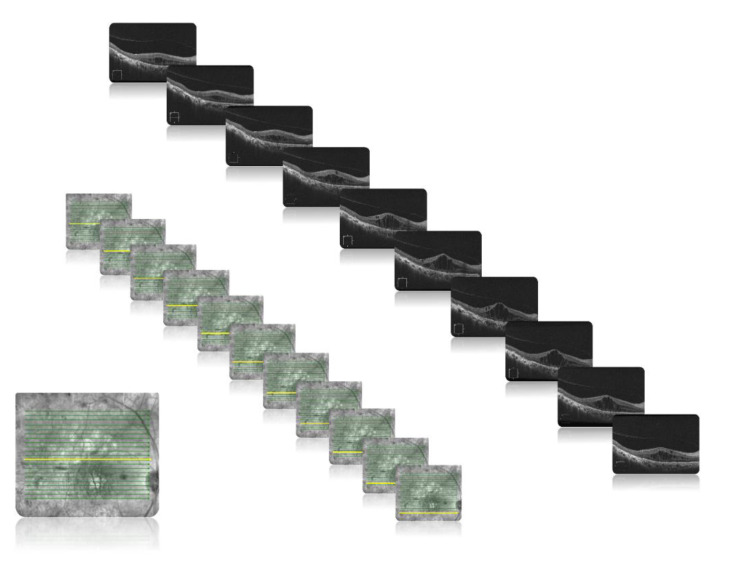
Example images from a 65-year-old man with cystoid macular edema. The OCT B-scans at different layers obtained from the same patient with macular edema were all enrolled in the study.

**Figure 2 jcm-11-02878-f002:**
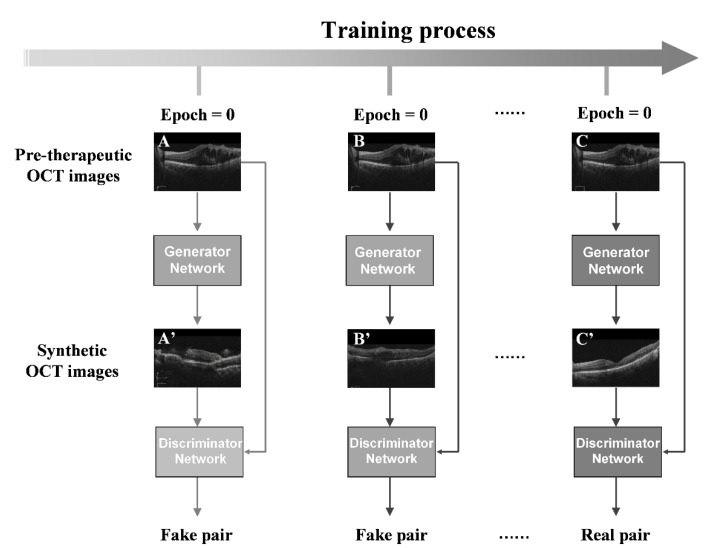
The training process of pix2pixHD. Illustration of the pix2pixHD-based solution used in this study for generating post-therapeutic OCT images from pre-therapeutic OCT images. The images (**A**–**C**) above represent the pre-operative OCT images. The images (**A**) in the left column show images from initial training that do not yet show retinal structures. The GAN models in the middle row have been trained to generate images (**B**) with retinal structures. The right column shows that the GAN model has been trained to generate a near-ground truth retinal image (**C**). GAN, generative adversarial network; OCT, optical coherence tomography.

**Figure 3 jcm-11-02878-f003:**
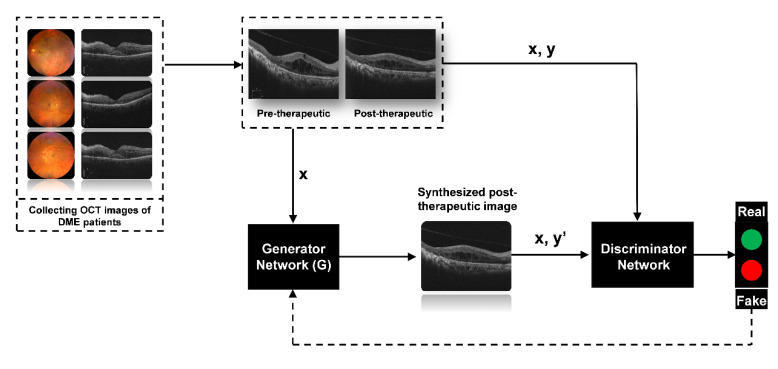
Illustration of generating post-therapeutic OCT from pre-therapeutic OCT by the GAN. OCT, optical coherence tomography; GAN, generative adversarial networks.

**Figure 4 jcm-11-02878-f004:**
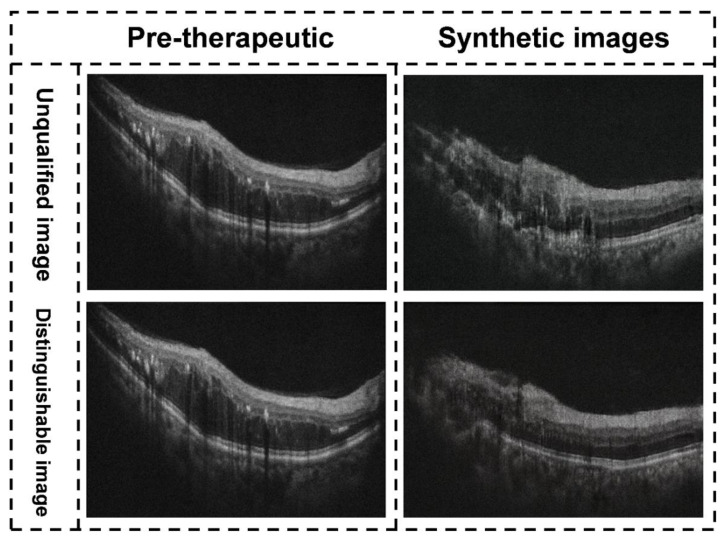
Examples of unqualified and distinguishable synthetic images. Illustration of the synthetic OCT images with different quality types. The images in the left column are pre-therapeutic images, and the images in the right column are synthetic post-therapeutic images by the pix2pixHD. The image on the upper right was considered inadequate by two retinal specialists during the evaluation process because it did not reflect the actual retinal structures; the image on the lower right was considered distinguishable by two retinal specialists during the evaluation process.

**Figure 5 jcm-11-02878-f005:**
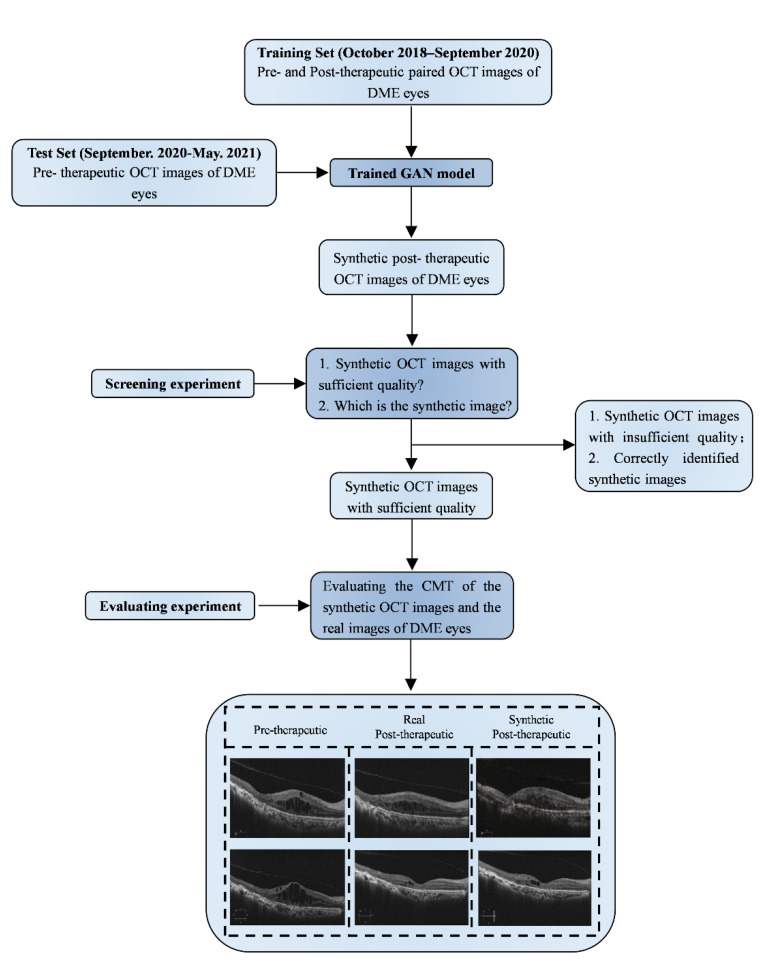
Workflow of our study. OCT, optical coherence tomography; DME, diabetic macular edema; GAN, generative adversarial networks; CMT, central macular edema.

**Figure 6 jcm-11-02878-f006:**
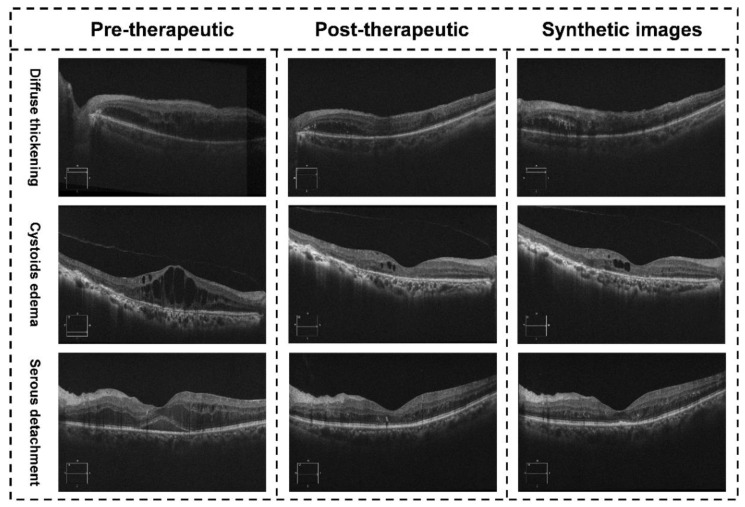
Illustration of the synthetic OCT images with different types of macular edema. The images in the right column are synthetic post-therapeutic images by the pix2pixHD. The images in the middle column are the actual image.

**Table 1 jcm-11-02878-t001:** Patient demographics.

	Training Set	Validation Set	*p*-Value
Patients (Female)	96 (47)	21 (11)	N/A
Ages	58.57 ± 9.14	56.57 ± 10.11	0.876
Eyes	107	26	N/A
Paired OCT images	561	71	N/A
VA baseline	0.581 ± 0.349	0.569 ± 0.316	0.651
VA 1-month	0.546 ± 0.313	0.524 ± 0.309	0.563
Injection phase			0.921
Loading phase	56 (52.33%)	15 (57.69%)	N/A
PRN phase	51 (47.67%)	11 (42.31%)	N/A
Anti-VEGF agent (%)			0.783
Ranibizumab	52 (48.60%)	12 (46.15%)	N/A
Conbercept	55 (51.40%)	14 (53.85%)	N/A
Classification of macular edema			N/A
Diffuse retinal thickening	87 (81.31%)	18 (69.23%)	0.103
Cystoids macular edema	79 (73.83%)	17 (65.38%)	0.328
Serous retinal detachment	23 (21.50%)	6 (23.08%)	0.823

VA, visual acuity, and the values are presented as the means ± standard deviations at baseline in different groups (in the logarithm of the minimum angle of resolution (logMAR) units).

**Table 2 jcm-11-02878-t002:** Comparisons of pix2pixHD against other state-of-the-art algorithms.

Algorithms	Unqualified Images Specialist #1	Unqualified Images Specialist #2	Identifiable Images Specialist #1	Identifiable Images Specialist #2
pix2pixHD	0	2	6	4
pix2pix	5	9	12	11
CRN	15	20	18	23

The images synthesized by three algorithms were judged by two retinal specialists, and comparisons of pix2pixHD against other state-of-the-art algorithms, pix2pix and CRN, are shown in detail.

**Table 3 jcm-11-02878-t003:** Accuracy of the Synthetic Post-therapeutic OCT Images of DME in the Evaluating Experiment.

CMT (μm)	Baseline	1-mo Prediction		
	Real Images	Synthetic Images	Real Images	MAE
Testing data	360.30 ± 224.34	330.35 ± 210.25	319.34 ± 208.65	24.51 ± 18.56
Injection phase				
Loading phase	365.43 ± 226.36	332.39 ± 214.87	320.67 ± 221.21	25.76 ± 20.25
PRN phase	354.67 ± 223.98	326.56 ± 209.32	319.76 ± 201.98	23.11 ± 17.79
Anti-VEGF agent (%)				
Ranibizumab	354.54 ± 219.46	340.34 ± 223.25	333.48 ± 221.09	26.78 ± 19.34
Conbercept	367.01 ± 228.37	323.18 ± 201.23	314.56 ± 203.39	22.39 ± 18.36
Classification of macular edema				
Diffuse retinal thickening	360.52 ± 225.37	335.39 ± 223.12	323.90 ± 215.91	22.11 ± 18.47
Cystoids macular edema	379.30 ± 238.78	329.59 ± 219.78	346.36 ± 238.85	32.45 ± 23.15
Serous retinal detachment	356.37 ± 227.74	327.35 ± 201.09	314.33 ± 197.64	23.87 ± 21.65

CMT, central macular thickness; PRN, pro re nata; MAE, mean absolute error, values are presented as the means ± standard deviations.

## Data Availability

Data was available from the corresponding authors for reasonable grounds.

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
