# Peer review of "Prediction of the Short-Term Therapeutic Effect of Anti-VEGF Therapy for Diabetic Macular Edema Using a Generative Adversarial Network with OCT Images"

_jcm, 2022, doi:10.3390/jcm11102878_

Round 1
Reviewer 1 Report
The study aims at using GAN model to generate OCT images to predict anti-VEGF therapy for DME. While this is an interesting study, the introduction can be expanded to include GAN modeling systems and their uses for OCT predictions thus far etc. Figure 1 is not clear and can be improved and labeled better. In figure 2, the pretherapeutic images look the same labeled under unqualified and distinguishable each. Are the meant to be the same image? Some grammatical errors can be corrected- such as line 284- some limitations should be considered not concerned. Images of more predictions can improve the paper.
Author Response
Reviewer #1
The study aims at using GAN model to generate OCT images to predict anti-VEGF therapy for DME. While this is an interesting study, the introduction can be expanded to include GAN modeling systems and their uses for OCT predictions thus far etc. Figure 1 is not clear and can be improved and labeled better. In figure 2, the pretherapeutic images look the same labeled under unqualified and distinguishable each. Are the meant to be the same image? Some grammatical errors can be corrected- such as line 284- some limitations should be considered not concerned. Images of more predictions can improve the paper.
Response: We are extremely grateful to the Reviewer for these pieces of advice.
- We added the introductions to generative adversarial networks and their uses for OCT predictions
Changes in the text (Lines 68-74, Page 2): A generative adversarial network (GAN) is an artificial intelligence-based "image-to-image" algorithm, which Ian Goodfellow first proposed in 2014, that can synthesize new images based on existing ones. The goal of GAN is to generate fake images that closely resemble the actual images. Over the past few years, the GAN-based algorithm has yielded satisfactory results when used to predict the effect of anti-VEGF or laser therapies for neovascular age-related macular degeneration (nAMD) and central serous chorioretinopathy (CSC).
- As suggested by the Reviewer, we reorganized the annotations in Figure 1 and revised the legend.
Changes in the text: Figure 1. The training process of pix2pixHD.
Illustration of the pix2pixHD-based solution used in this study for generating post-therapeutic OCT images from pretherapeutic OCT images. The images (A, B, C) above represent the pre-operative OCT images. The images (A and A') in the left column depict images from initial training that do not yet show retinal structures; GAN models in the middle row have been trained to generate images with retinal structures; The right column shows that the GAN model has been trained to generate a near-groundtruth retinal image. GAN, generative adversarial network; OCT, optical coherence tomography.
Changes in the text: Lines 131-137, Page 4.
- The pretherapeutic images in Figure 2 provided in the Supplemental materials were completely different from the unqualified and distinguishable images.
Previous studies about diabetic macular edema based on artificial intelligence, data forms including images and clinical data, etc., have substantiated that artificial intelligence technology is helpful for auxiliary diagnosis and prognosis prediction of DME. This study is aimed at the latter, and its effectiveness has been validated in follow-up experiments.
- The grammatical errors have been corrected by a native English-speaking ophthalmologist. Many thanks for your important and helpful suggestions on our manuscript entitled “Prediction of short-term therapeutic effect of anti-VEGF for diabetic macular edema using generative adversarial network with OCT images”. Based on your suggestions, we have carefully addressed all the issues and have modified our manuscript accordingly. Our references to the line numbers refer to the marked-up copy that we have uploaded as a ‘Revised Manuscript with Tracked Changes’ file. All the changes have been accepted in the clean revised manuscript uploaded as a ‘Manuscript’ file. We hope that we have adequately addressed your suggestions and that our manuscript is now suitable for publication. Please let us know if you have any further questions or suggestions.
Reviewer 2 Report
In this manuscript, the authors provide the results from the application of generative adversarial network (GAN) in predicting the outcome of anti-VEGF therapy for patients suffering from diabetic macular edema (DME). Their results suggest that for most cases the synthetic images were very similar to the actual post-operative OCT images, and for test cases the synthetic images generated by GAN were similar to the post-operative outcome. Thus, the results suggest that the pix2pixHD used by the authors can be used for predicting short-term outcome of anti-VEGF therapy in DME patients.
The authors make a good point about treatment compliance. Such prediction would help patients in following up clinical recommendations.
The following comments are meant to further improve the manuscript:
- Since the synthetic images would not replace the need for actual images, it would be good to mention post-operative images would still be needed.
- In this manuscript, the authors only used retrospective data. It would further enhance the manuscript if prospective data was included as well. If not, then it is somewhat misleading to include “Prediction of short-term therapeutic effect…..”. Without a prospective study the authors are not predicting yet using their system, but have established a system for prediction.
- “Most of the synthetic images (65/71) were difficult to accurately identify by retinal specialists.” Not clear what this means. Perhaps the authors mean that the specialists could not distinguish between synthetic and actual images, or could not tell whether an image was synthetic or not.
- Figures 1 and 2 have low resolutions images
- Figure 3 has low resolution text.
- For Figure 3 experimental flowchart for the manuscript, it is not clear what was done when the OCT images were of insufficient quality of if the image was correctly identified.
Author Response
Reviewer #2
In this manuscript, the authors provide the results from the application of generative adversarial network (GAN) in predicting the outcome of anti-VEGF therapy for patients suffering from diabetic macular edema (DME). Their results suggest that for most cases the synthetic images were very similar to the actual post-operative OCT images, and for test cases the synthetic images generated by GAN were similar to the post-operative outcome. Thus, the results suggest that the pix2pixHD used by the authors can be used for predicting short-term outcome of anti-VEGF therapy in DME patients.
The authors make a good point about treatment compliance. Such prediction would help patients in following up clinical recommendations.
The following comments are meant to further improve the manuscript:
- Since the synthetic images would not replace the need for actual images, it would be good to mention post-operative images would still be needed.
Response: We are extremely grateful for your expert advice. Indeed, even if predictive images are accurate, actual images are still needed to confirm treatment outcomes and assess macular edema status. We added this rigorous point to the discussion.
Changes in the text (Lines 362-364, Page 12): Last but not least, synthetic images cannot replace the need for actual images and are only used for prediction. Indeed, postoperative OCT images are still needed for clinical evaluation.
- In this manuscript, the authors only used retrospective data. It would further enhance the manuscript if prospective data was included as well. If not, then it is somewhat misleading to include "Prediction of short-term therapeutic effect…..". Without a prospective study the authors are not predicting yet using their system, but have established a system for prediction.
Response: We concur with the Reviewer. Indeed, it would further enhance the robustness of our predicting model if more and longer prospective data were included for analysis. This will be considered in future studies on establishing a long-term predicting model and verifying the predictive performance prospectively in the real world.
- "Most of the synthetic images (65/71) were difficult to accurately identify by retinal specialists." Not clear what this means. Perhaps the authors mean that the specialists could not distinguish between synthetic and actual images, or could not tell whether an image was synthetic or not.
Response: We are extremely grateful for this piece of advice.
Changes in the text (Lines 33-35, Page 1): Most synthetic images (65/71) were difficult to differentiate from the real OCT images by retinal specialists.
- Figures 1 and 2 have low resolutions images
Response: As suggested by the Reviewer, we generated TIFF images with a higher resolution for Figure 1 (3665×2896) and Figure 2 (7689×3493).
- Figure 3 has low resolution text.
Response: As suggested by the Reviewer, we generated TIFF images with a higher resolution for Figure 3 (4949×6563).
- For Figure 3 experimental flowchart for the manuscript, it is not clear what was done when the OCT images were of insufficient quality of if the image was correctly identified.
Response: We are extremely grateful for this comment. Handling images of insufficient quality can be challenging.
In our study, only synthetic images which were difficult to distinguish from original images of sufficient quality were further analyzed during the evaluation experiment. In other words, these images are discarded since when OCT images, such as eFigure 2, are generated, the small number of images of insufficient quality does not affect the overall judgment of the prediction.
Many thanks for your important and helpful suggestions on our manuscript entitled “Prediction of short-term therapeutic effect of anti-VEGF for diabetic macular edema using generative adversarial network with OCT images”. Based on your suggestions, we have carefully addressed all the issues and have modified our manuscript accordingly. Our references to the line numbers refer to the marked-up copy that we have uploaded as a ‘Revised Manuscript with Tracked Changes’ file. All the changes have been accepted in the clean revised manuscript uploaded as a ‘Manuscript’ file. We hope that we have adequately addressed your suggestions and that our manuscript is now suitable for publication. Please let us know if you have any further questions or suggestions.
Reviewer 3 Report
This study evaluated individualized post-therapeutic OCT images that could predict the short-term response of anti-VEGF therapy for DME using GAN. This study is somewhat interesting but has space to be improved to draw a reasonable and logical conclusion.
1. Because the algorithms in deep learning are affected by the imaging instruments, please provide the exact name of the OCT machine, scan mode, length of line scan, or scan area.
2. Please explain why one month was set to predict the therapeutic effect of anti-VEGF because one month is too short for determining the clinical effect of the anti-VEGF agent in most patients with DME.
3. Sample size is too small for the training and testing of the author’s model.
4. Quality of images is too poor to read the letters in the figures.
5. The inclusion and exclusion criteria for OCT images are unclear in this study. Please clarify how many images were excluded for specific reasons; finally, the 476 pairs of OCT images from 96 patients were assigned to the training data.
6. Please describe how the retinal specialists measure the CMT.
Author Response
This study evaluated individualized post-therapeutic OCT images that could predict the short-term response of anti-VEGF therapy for DME using GAN. This study is somewhat interesting but has space to be improved to draw a reasonable and logical conclusion.
- Because the algorithms in deep learning are affected by the imaging instruments, please provide the exact name of the OCT machine, scan mode, length of line scan, or scan area.
Response: We are extremely grateful to the Reviewer for this piece of advice. As suggested, we have added the exact name of the OCT machine (Carl Zeiss Meditec. Inc, Dublin, OH, USA) and the main scan mode HD 21-Line was used, which generates 21 high-definition horizontal scan lines at a depth of 2.0 mm with 8 acquired B-scans for each line and each B-scan composed of 1024 A-scans. The scan has an adjustable line length of 3, 6, or 9 mm. In the present study, we used a length of 9mm and a width of 6mm.
Changes in the text: Lines 116-124, Page 3.
- Please explain why one month was set to predict the therapeutic effect of anti-VEGF because one month is too short for determining the clinical effect of the anti-VEGF agent in most patients with DME.
Response: We thank the Reviewer for this insightful comment. Indeed, this is also a concern for many clinicians. In this study, we observed DME patients treated with 1+PRN regimens; patients that received one anti-VEGF injection during the first month, were observed monthly and received repeat injections based on OCT imaging. Indeed, not all patients responded satisfactorily to anti-VEGF injection. During clinical practice, switching to other potentially effective treatments at an early stage is recommended for patients with suboptimal responses. Accordingly, we predicted the short-term therapeutic effect at 1 month to evaluate the efficacy of the anti-VEGF therapy and determine the need to change the treatment strategy. It can be challenging to predict the therapeutic effect of macular edema in the second or third month since it is unclear whether patients will require repeated treatment or a change in anti-VEGF agent after the first month at the initial injection.
- Sample size is too small for the training and testing of the author's model.
Response: We concur with the Reviewer. Indeed, during clinical practice, the acquisition of large sample cohort data is often time-consuming and labor-intensive, and this can be challenging, especially during the pre-processing of retrospective datasets. Therefore, our team has recently started in-depth mining and utilizing small samples of high-quality data. To compensate for the limitation of insufficient sample size, we tried to extract many parameters of information from the OCT groups from a single examination. As shown in Figure 1, the OCT B-scan results at different layers obtained from the same patient with macular edema were all enrolled in the study. Furthermore, the results showed that our approach yielded good predictive performance. For further details, please refer to the performance of the GAN model in a large sample data set for comparison (Xu F, Wan C, Zhao L, Liu S, Hong J, Xiang Y, You Q, Zhou L, Li Z, Gong S, Zhu Y, Chen C, Zhang L, Gong Y, Li L, Li C, Zhang X, Guo C, Lai K, Huang C, Ting D, Lin H, Jin C. Predicting Post-Therapeutic Visual Acuity and OCT Images in Patients With Central Serous Chorioretinopathy by Artificial Intelligence. Front Bioeng Biotechnol. 2021 Nov 23;9:649221. doi: 10.3389/fbioe.2021.649221. PMID: 34888298; PMCID: PMC8650495.). We once again thank the Reviewer for this piece of advice. Although a small sample size yielded significant findings in this study, a larger sample is bound to yield a better model. We will continue to improve this part of the work in our future clinical studies.
- Quality of images is too poor to read the letters in the figures.
Response: We thank the Reviewer for this piece of advice. As suggested by the Reviewer, we have generated TIFF images with a higher resolution for Figure 1 (3665×2896), Figure 2 (7689×3493), Figure 3 (4949×6563) and Figure 4 (6230×4223).
- The inclusion and exclusion criteria for OCT images are unclear in this study. Please clarify how many images were excluded for specific reasons; finally, the 476 pairs of OCT images from 96 patients were assigned to the training data.
Response: Thank you for your advice. We modified the inclusion and exclusion criteria accordingly.
The inclusion criteria consisted of: (1) patients aged ≥ 18 years; (2) patients with a pre-operative diagnosis of DME based on fundus fluorescein angiography (FFA) and OCT; (3) patients treated with an injection of anti-VEGF therapy including conbercept or ranibizumab at any phase in the treatment protocol of three consecutive monthly injections and pro re nata (PRN) injections; and (4) OCT B-Scan results showing macular edema. The exclusion criteria were as follows: (1) presence of any other chorioretinal diseases, including age-related macular degeneration (AMD) and polypoidal choroidal vasculopathy (PCV); (2) low image quality caused by media opacities, or an abnormal signal strength index on the OCT images; and (3) OCT B-Scan results showing no macular edema in the 21-line mode pattern.
After careful calculation, a total of 2793 pairs of OCT images from 117 patients with 133 eyes were included in this study. Based on the above exclusion criteria, 2161 images could not be included, most of which were paired OCT images without macular edema from the periphery of a single examination with 21-line scan mode.
Changes in the text: Lines 89-102, Page 2-3.
- Please describe how the retinal specialists measure the CMT.
Response: We thank the Reviewer for this comment. We modified the manuscript and provided additional content on the measurement method of CMT.
CMT was measured as the outer surface of the line formed by the RPE to the outer surface of the retinal nerve fiber layer in the macular area. To ensure the consistency of measurement tools, the assessment of CMT of the synthetic images and the ground truth was counted manually using the ImageJ software (National Institutes of Health, Bethesda, MD, USA). However, given that measuring CMT is a very common operation for ophthalmologists, we only modified the manuscript based on your suggestion and did not add any pictures to the manuscript.
The yellow line represents the measurement range of the CMT.
Changes in the text: Lines 194-199, Page 6.
Many thanks for your important and helpful suggestions on our manuscript entitled “Prediction of short-term therapeutic effect of anti-VEGF for diabetic macular edema using generative adversarial network with OCT images”. Based on your suggestions, we have carefully addressed all the issues and have modified our manuscript accordingly. Our references to the line numbers refer to the marked-up copy that we have uploaded as a ‘Revised Manuscript with Tracked Changes’ file. All the changes have been accepted in the clean revised manuscript uploaded as a ‘Manuscript’ file. We hope that we have adequately addressed your suggestions and that our manuscript is now suitable for publication. Please let us know if you have any further questions or suggestions.
Round 2
Reviewer 1 Report
My questions have been adequately answered. Thanks